# Automaticity in processing spatial-numerical associations: Evidence from a perceptual orientation judgment task of Arabic digits in frames

**Shuyuan Yu[1], Baichen Li[2], Meng Zhang[3], Tianwei Gong[3], Xiaomei Li[4], Zhaojun Li[1], Xuefei Gao[5], Shudong Zhang[6]\*, Ting Jiang[3]\*, Chuansheng Chen[7]**

1 Department of Psychology, The Ohio State University, Columbus, Ohio, United States of America, 2 Department of Cognitive Neuroscience, Maastricht University, Maastricht, The Netherlands, 3 Falculty of Psychology, Beijing Normal University, Beijing, China, 4 Department of Human Development and Family Studies, University of Illinois at Urbana-Champaign, Champaign, Illinois, United States of America, 5 School of Psychology & Counselling, Queensland University of Technology, Brisbane, Australia, 6 Faculty of Education, Beijing Normal University, Beijing, China, 7 Department of Psychology and Social Behavior, University of California Irvine, Irvine, California, United States of America

☯ These authors contributed equally to this work.
\* psytingjiang@gmail.com (TJ); zsd@bnu.edu.cn (SZ)

## Abstract

Human adults are faster to respond to small/large numerals with their left/right hand when they judge the parity of numerals, which is known as the SNARC (spatial-numerical association of response codes) effect. It has been proposed that the size of the SNARC effect depends on response latencies. The current study introduced a perceptual orientation task, where participants were asked to judge the orientation of a digit or a frame surrounding the digit. The present study first confirmed the SNARC effect with native Chinese speakers (Experiment 1) using a parity task, and then examined whether the emergence and size of the SNARC effect depended on the response latencies (Experiments 2, 3, and 4) using a perceptual orientation judgment task. Our results suggested that (a) the automatic processing of response-related numerical-spatial information occurred with Chinese-speaking participants in the parity task; (b) the SNARC effect was also found when the task did not require semantic access; and (c) the size of the effect depended on the processing speed of the task-relevant dimension. Finally, we proposed an underlying mechanism to explain the SNARC effect in the perceptual orientation judgment task.

## Introduction

Research on numerical cognition has made considerable progress over the past decades [1–3]. One significant finding on the processing of magnitude is the automatic associations between numbers and space. In their seminal studies, Dehaene and his colleagues [4, 5] asked participants to judge the parity of the digits 0 to 9 by pressing left or right buttons, and found that

**Data Availability Statement:** All relevant data are within the paper and its Supporting Information files.

**Funding:** 2020 Comprehensive Discipline Construction Fund of Faculty of Education, Beijing

Normal University The sponsor play the role in the data collection and analysis.

**Competing interests:** The authors have declared that no competing interests exist.

participants were relatively faster to respond to small numbers with a left-sided response, and to large numbers with a right-sided reponse, which is known as the spatial-numerical association of response codes (SNARC) effect. The result suggests a number-space association, with small numbers associated with the left side and large numbers with the right side.

Since Dehaene et al.'s [4] initial study, researchers have used different tasks and stimulus materials to examine the SNARC effect. In addition to the *parity judgment task* [4, 6], some researchers have confirmed the effect with the *magnitude comparison task* [5, 7, 8], where participants are asked to judge whether a target digit is bigger or smaller than a reference number by pressing a left- or a right-hand key. Other researchers have investigated automatic numerical-spatial associations using *non-semantic tasks*, such as phoneme monitoring [6], color judgment [9, 10], orientation judgment [11, 12], and free viewing tasks [13]. In these tasks, magnitude information is less involved in task requirements than parity judgment or comparison tasks. For example, researchers observed a SNARC effect using a task where participants were asked to judge whether a digit was upright or tilted (10˚ to the right) [12], or to decide whether a triangle superimposed on a digit was pointing upward or downward [11]. Moreover, Fischer, Castel, Dodd, and Pratt [13] found that even when the digits were used merely as a fixation point, viewing small (or large) digits foster later decisions on targets on the left-side (or right-side) of the screen. Because these non-semantic tasks required participants to make judgments based merely on the perceptual attributes rather than semantic attributes of the stimuli, we refer to this kind of task as *perceptual judgment task* in this manuscript.

Using these different kinds of tasks (e.g., parity judgment, magnitude comparison, and perceptual judgment tasks), researchers are able to investigate the extent of automatic processing of numbers in a more nuanced way. Investigating automaticity helps us better understand the internal representations of numbers. According to Tzelgov and Ganor-Stern [14], automatic processes can be further distinguished as *intentional automatic processing*, where the process has to be part of the task requirements (e.g., SNARC effect observed in magnitude comparison tasks), and *autonomous automatic processing*, where the process occurs even when it is not part of the task requirements (e.g., SNARC effect observed in perceptual judgment tasks) [14–16]. Therefore, a stronger examination of the automaticity of numerical-spatial associations would be using perceptual judgment tasks.

Are there automatic numerical-spatial associations when magnitude information is task-irrelevant? Answers to this question seem to be inconsistent. Previous research showed that the SNARC effect in perceptual judgment tasks might depend on tasks [11] and stimulus modality [6, 9, 17], suggesting that automaticity might not be an all-or-none process, but on a more continuous spectrum [9].

In terms of task dependency, stronger SNARC effects are observed in orientation judgment tasks than color or shape judgment tasks [11]. Researchers observed a SNARC effect when participants judged whether a digit was upright or rotated [12], whether a triangle superimposed on a digit pointed upward or downward, and whether a line superimposed on a digit is horizontal or vertical [11]. However, there is no SNARC effect when participants judged whether a digit is red or green, or whether a shape superimposed on a digit is a circle or a square. Fias et al. [11] explained these results by neural overlapping between task-relevant and task-irrelevant processes. More specifically, semantic information of digits is known to be processed in the parietal cortex [18, 19]. When task-relevant processing also relies on the parietal cortex (e.g., orientation processing), task-irrelevant digit information is more likely to interfere with response time, whereas task-irrelevant digit information has little effect on response time when task-relevant processing minimally overlaps with the parietal cortex (e.g., color and shape processing).

Activation of magnitude information might also influence the SNARC effect through the response time. Wood and his colleagues [20] did a quantitative meta-analysis of 46 studies

with a total of 106 experiments differing in many aspects such as task, population, stimulus modality, stimulus format, and response modality. They found that the longer it took to respond, the larger the SNARC effect was. Similarly, Gevers and colleagues [21] compared the SNARC effect observed from different reaction time bins using a parity judgment task and found that the SNARC effect became stronger with increasing reaction time. More direct manipulation of digit viewing time showed that, contrary to the results that there is a lack of numerical-spatial associations in color decision tasks [11, 12], there appeared to be a SNARC effect in color decision tasks if digits are presented in black shortly (e.g., for 200ms) before color onsets (e.g. blue or green) [9, 10]. In this setting, participants had more time processing digit information and might activate strong enough magnitude information to interfere with reaction time for task-relevant color decisions.

The dual-route model proposed by Gevers et al. [21] provided an account of the underlying mechanisms of the SNARC effect and help explain the seemingly inconsistent findings. The model posits that numbers are processed automatically in terms of their spatial codes and consciously based on the task instructions. The model consists of three layers. The bottom layer represents the mental number line [5] and consists of a number field (nodes coding for each number) and a standard field (nodes coding for task-dependent features). The middle layer receives input from both number field and standard field, and consists of a magnitude field (two nodes coding for small and large magnitude) and additional fields that can be activated by the task requirements (e.g., two nodes for parity filed, one for odd and one for even). The magnitude and task-relevant fields will be activated in parallel. Finally, the top layer receives input from both magnitude field and task-relevant field and consists of nodes for left and right responses connected by lateral inhibition. Once a threshold is reached in one of the nodes in the top layer, a response is initiated.

With the assumption of parallel processing of magnitude and task-relevant information, the dual-route model explains the finding that the longer it takes to generate a motor response, the stronger is the impact of number magnitude on the response, and thus the stronger is the SNARC effect. What is more, the dual-route model can also explain other findings such as the categorically distributed SNARC effect in magnitude comparison tasks [5, 7, 8, 22], that is, the SNARC effect is stronger for numbers that are close to the standard (i.e., smaller distance) than for those that are farther apart (i.e., larger distance). This can be explained by the longer response time to close numbers [23].

However, most previous studies that explored the effect of activation of magnitude information through response time were based on either comparison across different studies, different tasks [20] or different participants [21], which are subject to sample biases. In the current study, we aimed to 1) further investigate the extent of automaticity in spatial-numerical associations between intentionally automatic processing (Experiment 1: parity judgment task) and autonomous automatic processing (Experiment 2–4: orientation judgment task); and 2) directly explore whether reaction time for task-relevant dimensions (e.g., orientation) influences task-irrelevant numerical-spatial associations in a within-subject design; and thus further examine the dual-route model. To achieve this aim, we used an orientation judgment task and systematically changed task difficulty (i.e., rotation degree) to manipulate response time under the same task instructions in a within-subject design (Experiment 2 and Experiment 4).

More specifically, in Experiment 1, we used a parity task to 1) provide a point of comparison for the SNARC effect in intentionally automatic processes. In the parity judgment task, the task-relevant parity judgment might itself activate magnitude information, and 2) replicate the SNARC effect in Chinese-speaking participants [24, 25]. Previous studies have demonstrated the SNARC effect with Chinese-speaking participants. Regardless of whether the participants

were readers of predominantly vertical texts (from top to bottom) [24] or readers of predominantly horizontal texts (from left to right) [25], they showed mappings of Arabic numbers onto a horizontal left-to-right number line.

In Experiments 2 to 4, we used an orientation task adapted from Lammertyn et al.'s [12] to investigate 1) to what extent magnitude information and its spatial associations are automatically accessed when magnitude information is task-irrelevant, and 2) the underlying mechanisms of the interaction between task-relevant and task-irrelevant information processing. In our tasks, the participants were asked to judge the orientation of a rotated digit (Experiment 2) or the frame surrounding a rotated digit (Experiments 3 and 4). In this paradigm, we were able to manipulate the difficulty of the task by changing the rotational angles of the Arabic digits. Based on the dual-route model, we hypothesized that when task is the most difficult (i.e., the rotated degree is the smallest), it would take longer time to process task-relevant information (i.e., to make an orientation decision), and thus it is more likely for the task-irrelevant information (i.e., magnitude information) to interfere with response time, indicated by a stronger SNARC effect.

## Experiment 1

In Experiment 1, we aimed to replicate the SNARC effect using a parity judgment task with Chinese-speaking individuals who predominantly read horizontal texts from left to right. The results also provided a point of comparison for the orientation judgment tasks in Experiments 2, 3, and 4.

### Method

**Participants.**   Thirty-two Chinese-speaking students (18 male and 14 female with a mean age of 22.19 years) at Beijing Normal University participated in Experiment 1. The participants gave written informed consent before taking part in the experiment and received a compensation of 10 RMB. All participants were right-handed and reported having a normal or corrected-to-normal vision. All participants have sufficient experience with Arabic digits. This experiment and all the following experiments in this paper were approved by the institutional review board of Beijing Normal University.

**Stimuli and procedure.**   The experiment was conducted in a behavioral laboratory with 3 Dell PCs with Tongfang 1775F Color Display Monitors (17 inches, resolution 1024*768). The experiment was programmed in E-Prime 1.0. The distance between the participant and the computer screen was approximately 30 cm.

Participants were presented with a number ranging from 0 to 9 and were asked to press the "F" key in response to an even number and the "J" key in response to an odd number. For each block, a random list of the numbers 0–9 was created and each number was repeated 10 times. No more than four stimuli with the same parity or two of the same stimuli were presented successively. Each block consisted of a total of 100 trials and each participant needed to complete two blocks of counterbalanced assignment of response keys.

In each trial, a white circle (2.1˚ in view) appeared for 500 ms as a fixation cue. The interval following the cue was a randomly timed (400–600 ms, mean = 500 ms) black screen, which helped to decrease the likelihood of a premature response. After the black screen, an Arabic number appeared (Arial font point size 64, 2.1˚ horizontally and 4.2˚ vertically in view), and remained on the screen for 150 ms. Participants were asked to decide the parity of each number and press the corresponding key. RTs were defined as the time from the onset of the digit to the onset of the key response. Each stimulus was presented centrally on a black background and the experiment would not continue to the next trial until a response was received. An

interval of 1000 ms was set between participants' response and the appearance of the fixation in the next trial. Each participant completed the experiment independently. It took approximately ten minutes to complete the task.

**Data analysis.** Data analysis was conducted with SPSS 20.0 [26]. Participants with a mean error rate greater than 20% of any hand were excluded from the final analysis. With data from one participant excluded, we had a final sample of 31 individuals whose mean accuracy was 94.7% ($SD$ = 4.0%).

Over the past decades, the classical way of analyzing the SNARC effect were regression analysis methods [6, 27]. Individual RT differences (dRT) for each number was computed by subtracting mean RT of left-sided responses from mean RT of right-sided responses. Then the regression analysis of dRT on magnitude of individual numbers would be conducted to measure the size of the SNARC effect. The negative regression slopes indicate the SNARC effect in the expected direction, i.e., faster left-sided (right-sided) responses for small (large) numbers. However, criticism of only using regression analysis to measure the SNARC effect was that although slopes reflect the linear relation between numbers and dRT, the effect size cannot be estimated in terms of proportion of variance explained [28, 29]. Thus, a repeated measures ANOVA of dRT with magnitude as a within-subject factor was suggested by Pinhas and Tzelgov and colleagues [28–30].

In the current study, we conducted our analysis in two ways. First, we conducted a repeated measures ANOVA of dRT with magnitude as a within-subject factor. In this approach, the SNARC effect would be revealed by a significant main effect of magnitude associated with a significant linear trend [28–30]. The effect size of the SNARC effect was denoted as the effect of the linear trend. Additionally, we conducted a regression analysis on dRT following Fias et al. [6] to compare our results with previously published SNARC studies. SNARC slopes can also give us a better understanding of the interaction between magnitude and response hand.

## Results

For the RT analysis, we excluded trials with wrong responses (9.1% of all trials) and RT more than 1500ms (.5%). The mean RT of the remaining trials was 510 ms ($SD$ = 81 ms). To avoid any potential bias of parity status on lateralized RT (i.e., the Markedness Association of Response Codes effect- MARC) [31], Tzelgov and colleagues proposed to use magnitude (small, intermediate, large) as the predictor of dRT ($RT_{Right}$−$RT_{Left}$) instead of numbers per se (1, 2, 3. . .) [28, 29]. Thus, we collapsed RT to an even and an adjacent odd number for each response hand and subject, resulting in five categories: very small (0, 1), small (2, 3), intermediate (4, 5), large (6, 7), and very large (8, 9). Then we computed dRT for each magnitude category and each subject. The repeated measures ANOVA on dRT with magnitude (very small, small, intermediate, large, very large) as within-subject factor showed a significant main effect of magnitude, $F$(4, 120) = 14.00, $p$ < .001, $\eta_p^2$ = .318. A trend analysis revealed a significant linear trend, $F$(1, 30) = 29.27, $p$ < .001, $\eta_p^2$ = .494, indicating a significant SNARC effect.

We also conducted a regression analysis of individual numbers on dRT following Fias et al. [6], which revealed a significant negative slope (unstandardized), $B$ = −9.05, one-tailed t comparison of $B$ with zero, $t$(30) = −4.88, $p$ < .001 (Fig 1).

## Discussion

Experiment 1 replicated the results of Dehaene et al. [4] by revealing the SNARC effect in a parity judgment task. This finding was also consistent with previous research with Chinese-speaking participants [24, 25]. Because the parity judgment task does not allow for easy manipulation

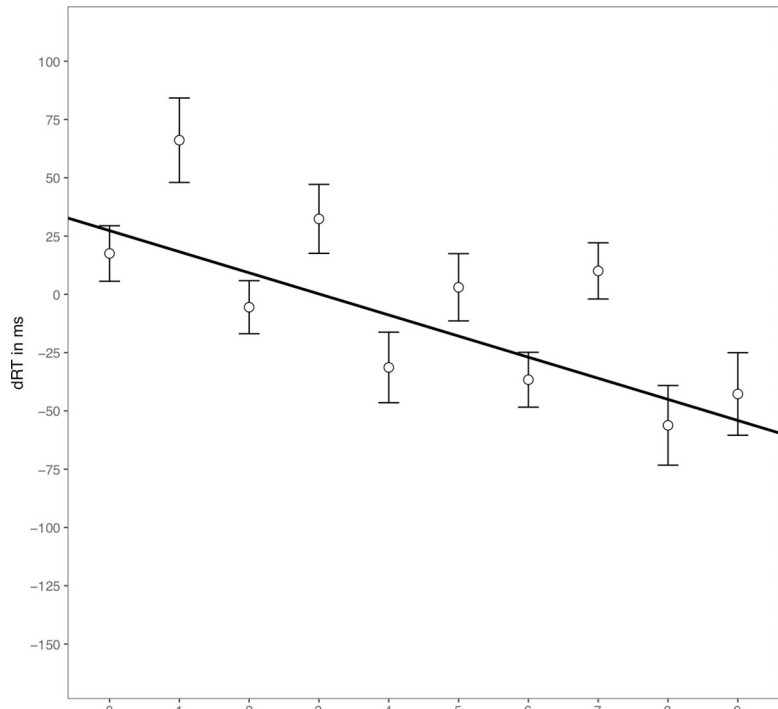

**Fig 1. Regression analysis of dRT ($RT_{Right} - RT_{Left}$) on magnitude category in Experiment 1.** Scattered dots indicate mean dRT by number. Error bars indicate standard errors. The continuous line indicates predicted dRTs based on regression analysis.

of task difficulty, we used an orientation judgment task in the subsequent experiments to further explore the underlying mechanisms of automatic numerical-spatial associations.

## Experiment 2

The objective of Experiment 2 was to compare the size of the SNARC effect at different levels of difficulty in a perceptual orientation judgment task. In this task, participants were asked to judge whether each rotated digit was clockwise or counterclockwise. We used the rotation of digits (3˚, 6˚, and 12˚ corresponding to Hard, Medium, and Easy) to manipulate the difficulty level of the task (and hence the processing speed for the task-relevant dimension).

### Method

**Participants.** Thirty-seven Chinese-speaking students (10 male and 27 female with a mean age of 21.16 years) at Beijing Normal University participated in Experiment 2. The participants gave written informed consent before taking part in the experiment and received a compensation of 20 RMB. All participants were right-handed and had a normal or corrected-to-normal vision. All participants have sufficient experience with Arabic digits.

**Stimuli and procedure.** Apparatus and experiment settings were the same as in Experiment 1. For the stimulus, a random list of the numbers 0–9 as the target stimuli was created, and the digits were rotated 3, 6, and 12 degrees for the Hard, Medium, and Easy levels of task difficulty respectively, resulting in each number having six possible orientations (left-3˚, 6˚ or 12˚ vs. right-3˚, 6˚ or 12˚). Each combination of number and orientation was repeated 10 times, resulting in a total of 600 trials. No more than four stimuli with the same rotation or two stimuli with the same number and rotation were presented successively.

The procedure for Experiment 2 was the same as that of Experiment 1 with the exception that participants were required to decide whether each number was left- or right-oriented, and press the left button ("F" on the keyboard) for the "left-oriented" (i.e., counterclockwise) stimuli and the right button ("J" on the keyboard) for the "right-oriented" (i.e., clockwise) stimuli. Each participant completed the experiment in no more than 25 minutes.

**Data analysis.** Data analysis was conducted with SPSS 20.0 [26]. Participants with a mean error rate more than 20% in any level of difficulty were excluded from the final analysis. With data from five participants excluded, we had a final sample of 32 individuals.

## Results

For the RT analysis, we excluded trials with incorrect responses (4.3%) or RT more than 1000 ms (2.4%). Mean accuracy and RT for each difficulty level are reported in Table 1. RTs were significantly different across Hard ($M = 520$ ms), Medium ($M = 465$ ms) and Easy ($M = 437$ ms) levels, $F(2,62) = 307.01$, $p < .001$, $\eta_p^2 = .908$, indicating an effective manipulation of difficulty.

We conducted a 3 (difficulty: Hard, Medium, Easy) * 10 (magnitude: 0, 1, 2, 3, 4, 5, 6, 7, 8, 9] repeated measures ANOVA of dRT with difficulty and magnitude as within-subject factors. The main effect of magnitude was significant, $F(9,279) = 39.71$, $p < .001$, $\eta_p^2 = .562$, associated with a significant linear trend, $F(1,31) = 47.78$, $p < .001$, $\eta_p^2 = .606$, which indicates an overall SNARC effect. The significant interaction effect between difficulty and magnitude confirmed our hypothesis that the SNARC effect would differ by difficulty level, $F(18, 558) = 13.17$, $p < .001$, $\eta_p^2 = .298$. Evaluating different difficulty levels separately, there was a significant SNARC effect in all levels of difficulty (Hard condition: main effect of magnitude $F(9,279) = 32.41$, $p < .001$, $\eta_p^2 = .511$; associated linear trend $F(1,31) = 20.75$, $p < .001$, $\eta_p^2 = .401$; Medium condition: main effect of magnitude $F(9,279) = 22.17$, $p < .001$, $\eta_p^2 = .417$; associated linear trend $F(1,31) = 35.82$, $p < .001$, $\eta_p^2 = .536$; Easy condition: main effect of magnitude $F(9,279) = 3.01$, $p = .002$, $\eta_p^2 = .088$; associated linear trend $F(1,31) = 4.92$, $p = .034$, $\eta_p^2 = .137$).

We additionally analyzed our data following Fias et al.'s [6] method to allow for comparisons with previously published SNARC studies. The regression analysis of individual digits on dRT revealed significant negative slopes (unstandardized) for all levels of difficulty (Fig 2). The Hard condition: $B = -6.21$, one-tailed comparison of $B$ with zero, $t(31) = -4.56$, $p < .001$; the Medium condition: $B = -5.31$, $t(31) = -5.99$, $p < .001$; the Easy condition: $B = -1.58$, $t(31) = -2.22$, $p = .017$. Furthermore, regression slopes differed across the three levels of difficulty, $F(2,62) = 6.18$, $p = .004$, $\eta^2 = .166$. Pairwise comparisons revealed that regression slopes in the Hard condition were significantly more negative than those in the Easy condition, $p = .007$; regression slopes in the Medium condition were significantly more negative than those in the Easy condition, $p = .009$; but the regression slopes were not different between the Hard and Medium conditions.

Finally, to compare the size of the SNARC effect in the parity judgment task (Experiment 1) and the numeral orientation judgment task (Experiment 2), we conducted a two-sample t-test between the regression slopes in Experiment 1 and the average regression slopes across three difficulty levels in Experiment 2. Results showed that slopes in Experiment 1 are significantly more negative than the slopes in Experiment 2, $t(36.9) = 2.39$, $p = .01$, indicating a stronger SNARC effect in a parity judgment task than a perceptual judgment task.

**Table 1. Mean proportion accuracy and RT (and standard deviations) for each difficulty level.**

|  | Easy | Medium | Hard | Overall |
|---|---|---|---|---|
| Accuracy | 98.6% (1.3%) | 97.5% (2.1%) | 93.4% (4.4%) | 96.5% (2.4%) |
| RT (ms) | 437 (59) | 465 (65) | 520 (75) | 474 (66) |

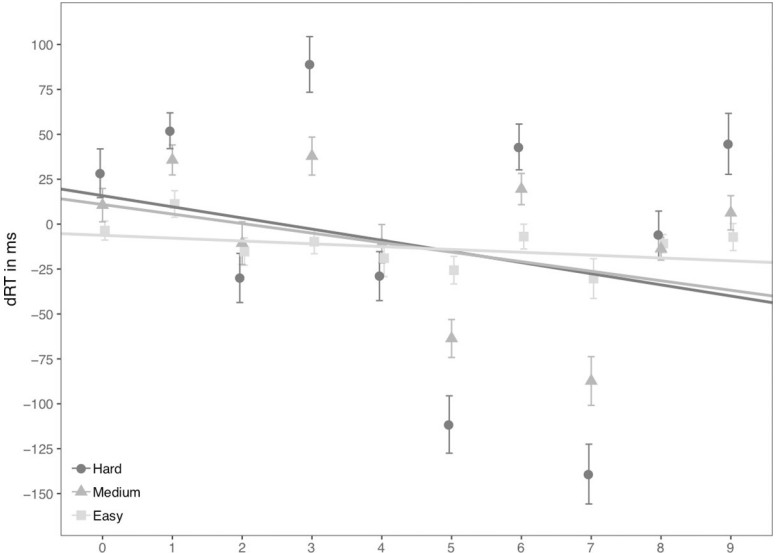

**Fig 2. dRT (RT$_{Right}$−RT$_{Left}$) for each number in Experiment 2.** Scattered dots indicate average dRT by number and difficulty level. Error bars indicate standard errors. Lines indicate predicted dRTs for three difficulty levels based on magnitude categories.

## Discussion

In Experiment 2 we found significant SNARC effects in the perceptual orientation judgment task across all three levels of task difficulty. The manipulation of task difficulty was effective as shown by the fastest response in the Easy condition, slower in the Medium condition, and the slowest in the Hard condition. Furthermore, there was a general trend that the SNARC effect became larger when task difficulty increased.

The results indicated that there were automatic spatial-numerical associations even when magnitude information was task-irrelevant. Moreover, the SNARC effect was smaller in the Easy condition than in the Medium and Hard conditions, suggesting that, at least within a certain range of difficulty levels, the longer it took participants to process task-relevant information (i.e., rotation), the stronger was the effect of the automatically activated task-irrelevant information (i.e., spatial-numerical associations). Furthermore, the SNARC effect elicited in the numeral orientation task was weaker compared to the parity task, indicating that the activation strength of magnitude information is stronger in intentionally automatic processes than autonomous automatic processes.

However, there was a potential confound in this task design. The perceptual characters of each Arabic digit might have led to different sub-levels of difficulty for different digits, as indicated by the significant main effect of number on RT, F(9, 279) = 7.10, $p < .001$, $\eta_p^2 = .186$. For instance, the rotated digit *1* could be easier to define its orientation than *3* in the same rotational degree because the rotation status of straight lines might be easier to clarify than that of curved lines, thus helping the overall performance of *1* over *3*. Experiment 3 overcame this problem with a modified perceptual judgment task.

## Experiment 3

In order to avoid inter-number perceptual variations and weak numerical-spatial associations with 0, we modified the orientation judgment task for Experiment 3 by adding a square-shaped frame outside the number that rotated the same degree as the number. Participants

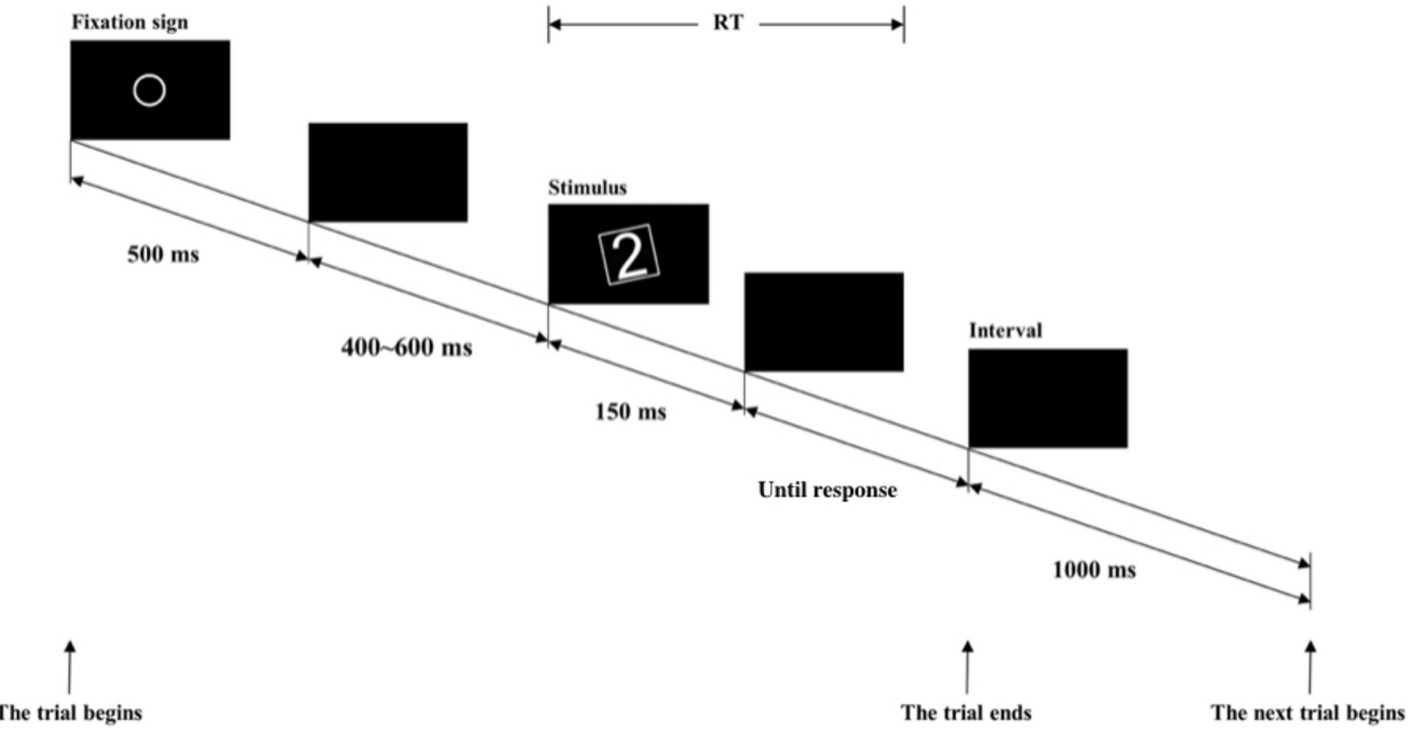

**Fig 3. Trial sequence and an example of the stimulus used in Experiment 3.**

were asked to judge the orientation of the rotated frame. Furthermore, we used only numbers 1–9 as stimuli in order to exclude the possible confusion of number 0. We refer to the modified task as the *frame* orientation judgment task (Fig 3) and the one used in Experiment 2 as the *numeral* orientation judgment task. We used only one level of difficulty (3˚, Hard) in Experiment 3 to examine whether there was a SNARC effect in this paradigm.

## Method

**Participants.** Thirty-seven Chinese-speaking students (6 male and 31 female with a mean age of 20.45 years) at Beijing Normal University participated in Experiment 3. The participants gave written informed consent before taking part in the experiment and received a compensation of 10 RMB. All participants were right-handed and had a normal or corrected-to-normal vision. All participants have sufficient experience with Arabic digits.

**Stimuli and procedure.** The stimuli and procedure for Experiment 3 was the same as those in Experiment 2, with three exceptions: (1) a square frame (each side was 4.7˚ in view) was added around the digit and participants were asked to judge whether the orientation of the frame was left-or-right-rotated (Fig 3); (2) there was only one difficulty level (3˚, Hard); and (3) the numbers as stimuli were restricted to 1–9. The experiment was programmed using Matlab2013b with PsychToolBox [32–34]. Each participant completed a total of 180 trials in 10 minutes or less.

**Data analysis.** Data analysis was conducted with SPSS 20.0 [26]. Participants with a mean error rate more than 20% were excluded from the final analysis. With data from one participant excluded, we had a final sample of 36 individuals whose mean accuracy was 93.9% ($SD = 3.7\%$).

## Results

For the RT analysis, we excluded trials with incorrect responses (6.7%) or RT more than 1000 ms (2.3%). The mean RT was 483 ms ($SD$ = 14 ms). First, we wanted to check whether by adding a frame we were able to control for the confound inter-number variations. There was no significant main effect of number on mean RT, $F(8, 280)$ = .73, $p$ = .669, suggesting that the control was effective.

Mean dRTs were subjected to repeated measures ANOVA with magnitude (1, 2, 3, 4, 5, 6, 7, 8, 9) as a within-subject factor. The main effect of magnitude was significant, $F(8, 280)$ = 2.72, $p$ = .007, $\eta_p^2$ = .072, however, the associated linear trend was not significant, $F(1, 35)$ = 1.31, $p$ = .260, indicating the absence of the SNARC effect. The regression analysis of dRT on digits revealed that the slopes (unstandardized) were not significantly different from zero, $B$ = −.94, one-tailed comparison of $B$ with zero, $t(35)$ = −1.15, $p$ = .130 (Fig 4).

## Discussion

Previous perceptual judgment tasks (including Experiment 2 in the current research) yielded different RTs for different numbers [4, 6, 11, 12]. By using a frame orientation judgment task, we controlled for that confound. We did not observe a SNARC effect in this experiment, possibly because here participants only need to focus on the rotated frame surrounding the digit, thus magnitude information is less activated than during the numeral orientation task in Experiment 2.

Moreover, there seems to be an absence of SNARC effect among large numbers (i.e., 7–9). We noticed a lack of the SNARC effect among the large numbers, which might cause the regression coefficient to be close to zero, and can also be observed in some other similar tasks [11, 12]. A potential explanation for this trend is that the response-related origin or the numerical-spatial associations responsible for the SNARC effect might be processed unevenly in digits 1–9 (stronger numerical-spatial associations in smaller range). Indeed, the numerical-spatial associations are observed to be stronger in the range 1–4 than the range 6–9 [35]. Therefore, in Experiment 4, we examine the SNARC effect using the same task in a smaller number range.

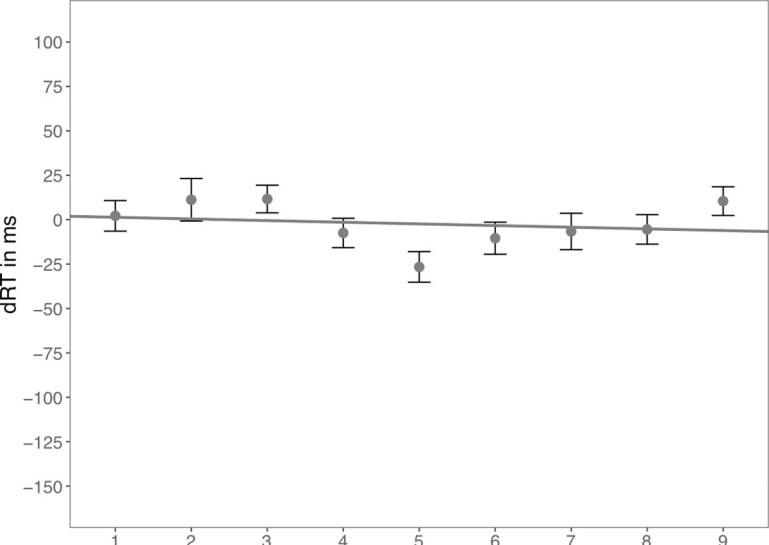

**Fig 4. dRT ($RT_{Right}$−$RT_{Left}$) for each number in Experiment 3.** Scattered dots indicate average dRT by number and difficulty level. Error bars indicate standard errors. Lines indicate predicted dRTs based on magnitude categories.

## Experiment 4

In Experiment 4, we aimed to further explore the effect of task difficulty on automatic associations of space and numbers in the frame orientation task using a smaller number range (i.e., 1–6) as an exploration.

### Method

**Participants.**   Twenty Chinese-speaking students (11 male and 9 female with a mean age of 22.85 years) from Beijing Normal University participated in Experiment 4. The participants gave written informed consent before taking part in the experiment and received a compensation of 20 RMB. All participants were right-handed and had a normal or corrected-to-normal vision. All participants have sufficient experience with Arabic digits.

**Stimuli and procedure.**   The stimuli and procedure for Experiment 4 were the same as those used in Experiment 3, except that in Experiment 4 number stimuli were restricted to 1–6, and that the stimuli were rotated 3, 6, and 12 degrees. Participants were asked to judge whether the orientation of the frame was left- or right-rotated. The experiment was programmed using 2013b with PsychToolBox [32–34]. Each participant completed a total of 360 trials in 15 minutes or less.

**Data analysis.**   Data analysis was conducted with SPSS 20.0 [26]. Participants with a mean error rate more than 20% in any level of difficulty were excluded from the final analysis. With data from one participant excluded, we had a final sample of 19 participants.

### Results

For the RT analysis, we excluded trials with incorrect responses (5.4%) or RT more than 1000 ms (.7%). Mean accuracy and RT for each difficulty level are reported in Table 2. There were significant differences across Hard ($M = 479$ ms), Medium ($M = 444$s) and Easy ($M = 422$ ms) levels, $F(2,36) = 90.40$, $p < .001$, $\eta_p^2 = .834$, which indicated that the manipulation of difficulty level was successful.

We then computed dRT ($RT_{Left}$—$RT_{Right}$) for each participant and each magnitude. A 3 (difficulty: Hard, Medium, Easy) * 6 (magnitude: 1, 2, 3, 4, 5, 6) repeated measures ANOVA on dRT with difficulty and magnitude as within-subject factors revealed a significant main effect of magnitude: $F(5,90) = 7.31$, $p < .001$, $\eta_p^2 = .289$. Trend analysis revealed a significant overall linear trend, $F(1,18) = 28.36$, $p < .001$, $\eta_p^2 = .612$, indicating an overall SNARC effect. The significant interaction effect between magnitude and difficulty level confirmed our hypothesis that the SNARC effect would differ by task difficulty level, $F(10,180) = 2.79$, $p = .003$, $\eta_p^2 = .134$. Separately analyzing the SNARC effect for each difficulty level, we observed significant SNARC effects in the Hard condition, but not the Medium and Easy conditions (Hard: the main effect of magnitude was significant, $F(5,90) = 8.91$, $p < .001$, $\eta_p^2 = .331$, associated linear trend $F(1,18) = 22.35$, $p < .001$, $\eta_p^2 = .554$; Medium: the main effect of magnitude was not significant, $F(5,90) = 1.95$, $p = .094$, but the associated linear trend was significant, $F(1,18) = 8.81$, $p = .008$, $\eta_p^2 = .329$; Easy: the main effect of magnitude was not significant, $F(5,90) = .79$, $p = .559$, associated linear trend was not significant, $F(1,18) = 2.89$, $p = .106$.

The regression analysis of dRT on digits following Fias et al. [6] revealed significant negative slopes in Hard and Medium conditions (Hard: $B = –12.50$, one tailed comparison of $B$

**Table 2. Mean proportion accuracy and RT (and standard deviations) for each difficulty level.**

|  | Easy | Medium | Hard | Overall |
|---|---|---|---|---|
| Accuracy | 97.9% (2.2%) | 95.7% (3.6%) | 92.1% (3.6%) | 95.2% (2.7%) |
| RT (ms) | 422 (56) | 444 (62) | 479 (70) | 448 (62) |

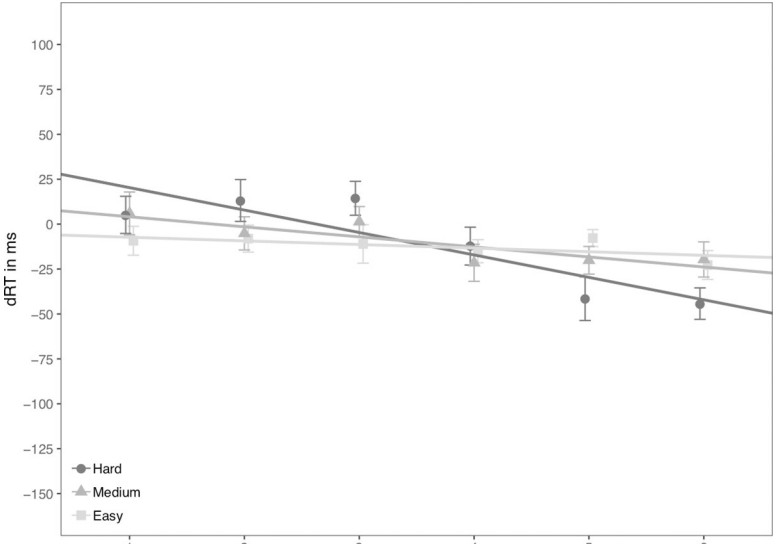

**Fig 5. dRT (RT$_{Right}$−RT$_{Left}$) for each number in Experiment 4.** Scattered dots indicate average dRT by number and difficulty level. Error bars indicate standard errors. Lines indicate predicted dRTs for three difficulty levels based on magnitude categories.

with zero, $t(18) = -4.73$, $p < .001$; Medium: $B = -5.60$, $t(18) = -2.97$, $p = .004$), but not in the Easy condition, $B = -2.02$, $t(18) = -1.70$, $p = .053$. Moreover, the slopes differed across three difficulty levels, $F(2,36) = 7.88$, $p = .001$, $\eta_p^2 = .304$. Pairwise comparison revealed that the regression slopes in the Hard condition were significantly more negative than those in the Medium ($p = .075$) and Easy conditions ($p = .006$), but the regression slopes were not significantly different between the Medium and Easy conditions ($p = .403$, see Fig 5).

## Discussion

Here we replicated our findings in Experiment 2 (numeral orientation judgment task) using a better-controlled task (frame orientation judgment task) with a smaller number range. The frame orientation judgment task in Experiment 4 revealed clear SNARC effects at the Hard level of task difficulty. It is worth noting that the significance of the SNARC effect in the Medium difficulty is inconsistent using an ANOVA analysis (lack of the main effect of magnitude on dRT) and regression analysis (significant negative slopes). However, in both analyses, the size of the SNARC effect becomes stronger as difficulty increases. This finding indicates that the longer it took to process the task-relevant dimension (orientation judgment), the greater was the impact of automatic processing of task-irrelevant magnitude information (the spatial numerical association), thus supporting the dual-route model [21].

Compared to the absence of the SNARC effect in the range 1–9 in Experiment 3, here we observed the presence of the SNARC effect in a smaller number range, which might be due to a clearer representation of relatively small numbers. Further discussions on the representations of number and associated space are presented in general discussion.

## General discussion

In the present study, we sought to examine whether the level of activation of magnitude information through task difficulty and response time affects the SNARC effect observed in non-sematic perceptual judgment tasks. To achieve this aim, we conducted four experiments using

different tasks: a parity judgment task (Experiment 1), a numeral orientation judgment task (Experiment 2), and a frame orientation judgment task (Experiments 3 and 4). A robust SNARC effect was detected in the parity judgment task in Chinese-speaking participants, the numeral orientation judgment task (across all three levels of task difficulty), and the frame orientation judgment task (for the Hard difficulty level and the 1–6 number range). More importantly, there was a clear tendency of larger SNARC effects as the difficulty level of the task increased, suggesting that the speed of processing of the task-relevant dimension influences the automatic processing of numerical-spatial associations.

Our results addressed two crucial research questions concerning visual number processing. First, does the automatic numerical-spatial processing occur when the task does not require semantic access? Second, does the impact of the automatic processes depend on the processing speed on the task-relevant dimension?

Research has shown that response time influences the size of the SNARC effect [20, 21]. More specifically, the longer time it takes to reach a response, the stronger the SNARC effect is. To explain this phenomenon, the dual-route model [21, 36] posits that the SNARC effect consists of a relatively fast unconditional route that automatically codes for magnitude information and the response-related spatial information of the stimulus and a relatively slow conditional route that is dependent on the task instructions and provides the mapping of the relevant attributes (e.g., magnitude, parity) to the required response. According to this model, the longer it takes to generate a motor response through the conditional route, the stronger the impact of automatic processing of magnitude information on the response through the unconditional route, thus the stronger the SNARC effect.

However, there are two limitations in the previous studies. First, most studies considered the effect of response time were based on comparisons across different studies or participants, for example, tasks or participants with longer response latencies are associated with larger SNARC effect, which are subjects to sample biases. Second, models explaining the effect of response time on the size of the SNARC effect mainly focus on semantic tasks (e.g, magnitude and parity judgment tasks) that involve working memory. However, a process (e.g., numerical-spatial associations) is more automatic if it can happen when it is not part of the task requirements [14], as in non-semantic perceptual judgment tasks (e.g., orientation judgment tasks). For example, Cleland and Bull [9] found that there is no SNARC effect in a color decision task (whether a digit was blue or green), however, a SNARC effect appeared when the digit was presented in black shortly (e.g., 200 ms) before the color onset. Their findings indicated a stronger SNARC effect as the viewing time of a digit increases. In other words, magnitude information is more likely to interfere with response time when the onset of task-irrelevant magnitude information is earlier than the onset of task-relevant color information, therefore supporting the dual-route account in autonomous automatic processing.

In the current study, we further examine the automaticity of numerical-spatial associations in an orientation task whether the task-relevant (orientation) and task-irrelevant (digits) information had the same onset. To investigate the effect of processing time of task-relevant information, we manipulated task difficulty by changing the rotated degrees in a within-participants design. Therefore, this manipulation might be a stronger examination of autonomous automatic processing of spatial-numerical associations than an extra viewing time of task-irrelevant digit information alone. Consistent with previous studies [9–13], we observed that automatic numerical-spatial processing occurred when the task requires non-semantic access. Moreover, we observed that the size of the SNARC effect is in general larger in the parity judgment task (Experiment 1) than perceptual judgment tasks (Experiment 2), supporting the account that the SNARC effect depends on the activation of magnitude information.

More importantly, we provided empirical evidence that the size of the SNARC effect was influenced by task difficulty in a non-semantic perceptual judgment task. Like the dual-route model [21], our results support a parallel-processing mechanism. Unlike the dual-route model, in perceptual tasks, the conditional route does not require the processing of magnitude information in working memory, because magnitude information was task-irrelevant. In other words, the task-relevant (e.g., perceptual features of the Arabic digits, such as the orientation of the frames) and the task-irrelevant information (e.g., numerical features of the Arabic digits) are processed along two independent pathways in parallel. As task-relevant dimensions (e.g., the orientation of the frame) became more difficult, it takes a longer time to generate a motor response, thus the task-irrelevant magnitude information has a stronger impact on response, yielding a stronger SNARC effect. This model for the SNARC effect in perceptual judgment tasks shares similar mechanism as the number Stroop effect (participants are asked to compare the physical size or numerical value of two numbers, and they respond faster when the physical and semantic dimensions are congruent than they are incongruent) [16, 37–38].

Because of the differences in perceptual and parity judgment tasks, we made two predictions for perceptual judgment tasks to be further tested. First, for parity judgment tasks, research has shown that the SNARC effect is notation-independent [39]. However, for perceptual judgment tasks, converging evidence suggests the SNARC effect depends on the modalities of stimuli. For example, researchers observed numerical-spatial associations with Arabic digits (i.e., 8), but not with non-symbolic numerosities (e.g., 8 circles), verbal words (e.g., eight), or auditory words (e.g., the sound of eight) [6, 9, 17, 35, 40]. Thus, we predicted that the SNARC effect for perceptual judgment tasks might be more sensitive to the notation of the magnitude than parity judgment tasks.

Second, for parity judgment tasks, the SNARC effect is modulated by the relative magnitudes within the tested interval [4]. Moreover, the working memory account posits that the SNARC effect is based on representations created in the serial order working memory [41, 42]. For instance, van Dijck and Fias [42] asked participants to hold a randomly ordered number sequence in the working memory during a parity task, and found a SNARC effect for ordinal positions of the number sequence instead of a SNARC effect for absolute magnitudes. However, here in the perceptual tasks, since the semantic information of magnitude is not necessarily activated in the working memory to solve the task, we hypothesized that the SNARC effect might not be affected by different intervals, but modulated only by the absolute magnitude. Future studies are needed to directly test these predictions.

Furthermore, the automatic numerical-spatial associations might provide an insight into the representation of magnitude. It is generally believed that the representation of nonsymbolic numerosities (e.g., 20 apples) becomes noisier as the number increases in an Approximate Number System (ANS) [43]. This is suggested by two main accounts: the linear model (linear representations of numbers with linearly increasing variability as magnitude increases) [44] and the log model (logarithmic representations of numbers with fixed variability around numbers) [45]. Consistent with these accounts, previous studies using non-symbolic numerosities (1–9 triangles) in an orientation decision task also revealed a stronger SNARC in 1–4 compared to 6–9, indicated a more precise spatial association in a smaller number range [35].

As for Arabic digits, educated adults are able to represent numbers linearly [46]. A common task to measure the representation of numbers is the number line task [46] (e.g., where is 345 on a 0–1000 number line?), where intentional processing of magnitude information is required. However, in the current study, we observed potentially stronger spatial-numerical associations in a smaller number range (1–6) than a larger number range (1–9) using a task where magnitude information is task-irrelevant (Experiment 3 and Experiment 4). A potential

explanation is that there might be more noise in representations and a less associated spatial precision in the larger range when magnitude information is weakly activated.

To conclude, we provided evidence that indicates automatic processing of numerical-spatial associations using a non-semantic perceptual judgment task. Moreover, the visibility of the automatic processes depends on the processing speed of the task-relevant dimension, indicating a dual-route processing mechanism. Further studies need to be conducted to investigate potential underlying mechanisms of the SNARC effect in the perceptual judgment tasks.

## Supporting information

**S1 Data. Data from Experiment 1.**
(CSV)

**S2 Data. Data from Experiment 2.**
(CSV)

**S3 Data. Data from Experiment 3.**
(CSV)

**S4 Data. Data from Experiment 4.**
(CSV)

**S1 Codebook. Codebook for data in Experiment 1–4.**
(CSV)

## Author Contributions

**Conceptualization:** Shudong Zhang, Ting Jiang.

**Formal analysis:** Tianwei Gong, Xiaomei Li, Zhaojun Li, Xuefei Gao, Shudong Zhang, Ting Jiang.

**Funding acquisition:** Shudong Zhang.

**Investigation:** Tianwei Gong, Xiaomei Li, Zhaojun Li, Xuefei Gao.

**Methodology:** Shudong Zhang, Ting Jiang.

**Software:** Shuyuan Yu, Baichen Li.

**Supervision:** Shudong Zhang, Ting Jiang.

**Visualization:** Tianwei Gong, Xiaomei Li, Zhaojun Li, Xuefei Gao.

**Writing – original draft:** Shuyuan Yu, Baichen Li, Meng Zhang.

**Writing – review & editing:** Shuyuan Yu, Shudong Zhang, Ting Jiang, Chuansheng Chen.

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
