## [Decision Letter · Decision Letter 0]

27 Oct 2019

PONE-D-19-16766

Automaticity in processing spatial-numerical associations: Evidence from a perceptual orientation judgment task of Arabic digits in frames

PLOS ONE

Dear Dr. Zhang,

Thank you for submitting your manuscript to PLOS ONE. After careful consideration, we feel that it has merit but does not fully meet PLOS ONE’s publication criteria as it currently stands. Therefore, we invite you to submit a revised version of the manuscript that addresses the points raised during the review process.

As you will see, the two reviewers have different views of the manuscript: one suggests minor revisions, one suggests rejection. I have read the manuscript myself, and I somehow agree with both reviewers: the manuscript has some merit, but the rationale for the experiments and the discussions of the findings need to be better framed within the existing literature. Also, the reviewers raised some issues regarding the analyses.

I decided to offer you the opportunity to revise the manuscript. If you will decide to revise the manuscript, please address all points raised by the reviewers. Moreover, please pay special attention to the analyses of the data and on how you report them. If you will resubmit the manuscript, I will try to get the same reviewers. 

We would appreciate receiving your revised manuscript by Dec 11 2019 11:59PM. To enhance the reproducibility of your results, we recommend that if applicable you deposit your laboratory protocols in protocols.io, where a protocol can be assigned its own identifier (DOI) such that it can be cited independently in the future. For instructions see: http://journals.plos.org/plosone/s/submission-guidelines#loc-laboratory-protocols

We look forward to receiving your revised manuscript.

Kind regards,

Claudio Mulatti, Ph.D.

Academic Editor

PLOS ONE

Journal Requirements:

2. Please provide additional details regarding participant consent. In the Methods section, please ensure that you have specified what type of consent you obtained (for instance, written or verbal). If verbal consent was obtained please state why it was not possible to obtain written consent and how verbal consent was recorded. If your study included minors, state whether you obtained consent from parents or guardians.

Reviewers' comments:

Reviewer's Responses to Questions

**Comments to the Author**

1. Is the manuscript technically sound, and do the data support the conclusions?

Reviewer #1: Partly

Reviewer #2: Yes

2. Has the statistical analysis been performed appropriately and rigorously? 

Reviewer #1: No

Reviewer #2: Yes

3. Have the authors made all data underlying the findings in their manuscript fully available?

Reviewer #1: Yes

Reviewer #2: Yes

4. Is the manuscript presented in an intelligible fashion and written in standard English?

Reviewer #1: Yes

Reviewer #2: Yes

5. Review Comments to the Author

Reviewer #1: In this paper, the authors report 4 experiments designed to investigate the automaticity of spatial-numerical associations. In the first study, they demonstrate a left-to-right SNARC effect for parity decisions in Chinese speakers. In the second study, participants perform a perceptual orientation judgment task where they judge whether rotated digits are rotated clockwise or counterclockwise at three levels of difficulty (3, 6, and 12 degrees rotation). They again report a SNARC effect with the effect tending to increase with difficulty. In Experiment 3 they present digits inside a square frame and ask participant to respond to the frame orientation rather than the digit orientation, but do not find a significant SNARC effect. In Experiment 4, they limit digits to 1-6 and find a SNARC effect in the same task but only for the hard and medium conditions. The authors conclude that they have found evidence for automatic processing of spatial-numerical associations using a non-semantic perceptual judgment but that the effect depends upon the processing speed of the task-relevant dimension. They interpret the findings in the context of Gevers et al. (2006) dual-route model.

The line of research certainly has potential and the findings from the orientation task are interesting, but I have some reservations about the paper that would make me reluctant to recommend it for publication as it stands. I also have some issues with the analysis (outlined further below).

I found the theoretical content somewhat sparse. How do the findings fit with Fias et al.’s (2001) suggestion that SNARC effects for orientation arise because of overlapping parietal processing for digits and orientation? I’d also be interested how the results fit with working memory accounts of SNARC (e.g., van Dijck & Fias, 2011). The paper is titled as examining the automaticity of SNARC effects, but there is comparatively little discussion of automaticity in the introduction or general discussion. A more nuanced discussion of what the authors mean by “automatic” would strengthen the paper considerably.

There was another recent paper that examined the automaticity of SNARC effects for perceptual judgment (in this case, color decision), and the authors might find it interesting as the pattern of results they report has some interesting parallels with the current paper (this paper reports that the SNARC effect does not arise for simple color decision, but does arise when participants either perform a go/no-go task, or view the digit for sufficient time before making their decision). The paper may be useful for the authors, particularly for the General Discussion (lines 394-404) when they talk about the need for non-semantic perceptual tasks. The findings are also consistent with one of the authors’ predictions (line 422), where they predict that perceptual judgment tasks might be sensitive to notation (in this paper, non-symbolic numerosities do not show the same pattern of effects as digits):

Cleland & Bull (2019). Automaticity of access to numerical magnitude and its spatial associations: the role of task and number representation, Journal of Experimental Psychology: Learning, Memory and Cognition, 45(2), 333-348.

I found it difficult to follow the rationale for including Experiment 1. I’m assuming the purpose is to first replicate the finding that Chinese speakers show a left-to-right SNARC effect for parity decision to digits before going onto the more perceptual tasks. However, this rationale needs to be explained in more detail in the Introduction, and the finding should be discussed in the General Discussion.

I have outlined further comments by line number below.

Line 66, [9] Keus et al. (2005) is cited as an example of a study that used color naming; however Keus et al. is an ERP study using parity decision so I think there is an error here. I am not aware of any study that uses color *naming* as a task, although there are several that use color decision although not many report a SNARC effect. The authors could cite Fias et al. (2001) or Lammertyn et al. (2002), although neither of these studies found a SNARC effect for color decision. Hoffmann et al. (2013) reported a SNARC effect for children with color decision when the digit was presented in black for 200 ms before changing color. Cleland and Bull (2019) reported a SNARC effect for color decision in adults, but only under certain conditions (see comment above).

Hoffmann, D., Hornung, C., Martin, R., & Schiltz, C. (2013). Developing number–space associations: SNARC effects using a color discrimination task in 5-year-olds. Journal of Experimental Child Psychology, 116(4), 775-791.

Line 110 – the participants are university students so I assume they are used to working with Arabic digits, but it may just be worth clarifying this to the reader (who may be wondering about their proficiency with Arabic digits)

Line 159 – talk the reader through how you ran the Fias et al. analysis. Was this based on binned responses as well?

Line 205 – I can follow the explanation for binning based on the MARC effect in Experiment 1, but do you still need to do this for the orientation tasks? As the participants are not performing a parity task, I don’t think you would expect to see a MARC effect. In particular, I can’t tell from the text whether you have binned the responses for the Fias et al. style analysis, but I’m not sure there’s a reason to do this if you have.

Line 216 – is there a reason not to report exact p-values? Unless it is journal policy, I’d recommend following APA guidelines and reporting exact p-values rather than <.01 or .05.

Line 277 – I’ve been trying to think through whether it matters that 1 is its own bin whereas all other bins have 2 numbers in them. I am not sure that this is a good idea – would it not make sense to abandon the bins here, given that you cannot have equal numbers of trials in each? Excluding 5 from your stimuli would have been one solution to this.

Line 289 onwards – I’m uncomfortable with the separate analysis of ranges 1-6 and 4-9. I can see no particular reason why you would predict that there would be a SNARC effect for 1-6 that then reversed for 4-9 (and I note that 4, 5, and 6 are included in both analyses). So why would you run this analysis? I can’t think of a better way of saying it than that this feels like a fishing expedition. There are many ways that you could have sliced up the data, and (unless you have pre-registered this somewhere) I don’t think there’s sufficient justification for this. This is why I've put "no" to the question about whether the analysis is rigorous.

Futhermore, if you are arguing that SNARC effects reverse for the higher number range, then this is a strong claim and needs to be returned to in the General Discussion and (potentially) replicated.

Line 304 – “discovered opposing SNARC effects for two number ranges” – I really don’t think you can say this. Firstly, you have sliced up the data without planning to originally. But also, I don’t think you can argue that you have two SNARC effects – the evidence just isn’t strong enough.

Line 313 – why are there so many fewer participants in this study than in the previous studies?

Reviewer #2: Overall Evaluation:

The paper is well written and I believe that the experiments operationalize very well the concepts that the authors present in the introduction. The experiments feel in very well a gap of knowledge that the discipline had, moreover confirming Gevers et al.’s (2006) model. I have a couple remarks before I can recommend the manuscript to be accepted. The remarks are listed below. The only main point is that the authors did note completely discuss the fact that they found a SNARC effect only for numbers that go from 1 to 5 in Experiment 3 (then replicated in Experiment 4 with the interval 1-6) and that they found a reverse SNARC effect for numbers that go from 6 to 9 in Experiment 3. My recommendation is to accept the manuscript with minor revision.

Line-by-line comment:

p. 4, l.93

I think “are” is missing in the middle of “which subject”

p. 5, l.107

The authors write: “The results also provided a point of comparison for the new task of orientation judgment for Experiments 2, 3, and 4.” However, the authors never compare the other experiments to experiment 1, so I am not sure it is really the purpose. I think that a better point of comparison, would have been an experiment with empty squares that are tilted clockwise or counterclockwise (I am not asking for the addition of a supplementary experiment)

p. 6, l.148

I would like to know on what ground the authors determined a cut-off at 1500ms?

p. 6, l.178

Why did the authors use 37 participants, what was the rational in terms of power of the analysis? I am asking because in the first experiment only 32 participants were used whereas in experiment 4, 20 were used.

p. 10, l.241

The authors write: “However, there was a potential confound in this task design. The perceptual characters of each Arabic digit might have led to different levels of difficulty, as indicated by the significant main effect of number on RT, F(9, 279) = 7.10, p < .001, ηp2 = .186. Experiment 3 overcame this problem with a modified perceptual judgment task.” Could they be more explicit, I am asking this because the digits (and therefore their perceptual characters) are manipulated orthogonally to the task difficulty, so I don’t see how there could be a confound?

p. 11, l.273

Why did the authors cut reaction times over 1000ms here (same in experiment 2) while cutting reaction times over 1500ms in experiment 1?

General Discussion

The general discussion is good but it does not seem (or maybe I missed it) to address the elephant in the room. Why is there a SNARC effect only for numbers that go from 1 to 5 in Experiment 3 and then replicated in Experiment 4? And why there seem to be a reverse SNARC effect for numbers that go from 6 to 9 in Experiment 3. The authors would need to address that.

6. PLOS authors have the option to publish the peer review history of their article (what does this mean?). If published, this will include your full peer review and any attached files.

Reviewer #1: No

Reviewer #2: No

---

## [Author Response · Author response to Decision Letter 0]

11 Dec 2019

Review Comments to the Author

Reviewer #1: In this paper, the authors report 4 experiments designed to investigate the automaticity of spatial-numerical associations. In the first study, they demonstrate a left-to-right SNARC effect for parity decisions in Chinese speakers. In the second study, participants perform a perceptual orientation judgment task where they judge whether rotated digits are rotated clockwise or counterclockwise at three levels of difficulty (3, 6, and 12 degrees rotation). They again report a SNARC effect with the effect tending to increase with difficulty. In Experiment 3 they present digits inside a square frame and ask participant to respond to the frame orientation rather than the digit orientation, but do not find a significant SNARC effect. In Experiment 4, they limit digits to 1-6 and find a SNARC effect in the same task but only for the hard and medium conditions. The authors conclude that they have found evidence for automatic processing of spatial-numerical associations using a non-semantic perceptual judgment but that the effect depends upon the processing speed of the task-relevant dimension. They interpret the findings in the context of Gevers et al. (2006) dual-route model.

The line of research certainly has potential and the findings from the orientation task are interesting, but I have some reservations about the paper that would make me reluctant to recommend it for publication as it stands. I also have some issues with the analysis (outlined further below).

I found the theoretical content somewhat sparse. How do the findings fit with Fias et al.’s (2001) suggestion that SNARC effects for orientation arise because of overlapping parietal processing for digits and orientation? I’d also be interested how the results fit with working memory accounts of SNARC (e.g., van Dijck & Fias, 2011). The paper is titled as examining the automaticity of SNARC effects, but there is comparatively little discussion of automaticity in the introduction or general discussion. A more nuanced discussion of what the authors mean by “automatic” would strengthen the paper considerably.

Response: Thank you so much for your comments on our studies. We agree that more theoretical content is needed and revised our introduction and discussion part to include more theoretical backgrounds. The responses for specific questions are as below.

How do the findings fit with Fias et al.’s (2001) suggestion that SNARC effects for orientation arise because of overlapping parietal processing for digits and orientation?

Response: Our findings that a SNARC effect was observed in a numeral orientation task (Experiment 2) can be explained by Fias et al.’s (2001) that the SNARC effects for orientation arise because of overlapping parietal processing for digits and orientation. However, the absence of the SNARC effect in the frame orientation task (Experiment 3) suggests that simply neural overlapping processing for digits and orientation might not be enough to elicit spatial-numerical associations. The strength of task-irrelevant magnitude information activation is also crucial.

p. 4, l. 90-101:

In terms of task dependency, stronger SNARC effects are observed in orientation judgment tasks than color or shape judgment tasks [11]. Researchers observed a SNARC effect when participants judged whether a digit was upright or rotated [12], whether a triangle superimposed on a digit pointed upward or downward, and whether a line superimposed on a digit is horizontal or vertical [11]. However, there is no SNARC effect when participants judged whether a digit is red or green, or whether a shape superimposed on a digit is a circle or a square. Fias et al. [11] explained these results by neural overlapping between task-relevant and task-irrelevant processes. More specifically, semantic information of digits is known to be processed in the parietal cortex [18, 19]. When task-relevant processing also relies on the parietal cortex (e.g., orientation processing), task-irrelevant digit information is more likely to interference with response time, whereas task-irrelevant digit information has little effect on response time when task-relevant processing minimally overlaps with the parietal cortex (e.g., color and shape processing).

How do the results fit with working memory accounts of SNARC (e.g., van Dijck & Fias, 2011)?

Response: The current results did not directly address whether the SNARC effect is based on serial order working memory, because the current design cannot distinguish magnitude information in the long-term memory and ordinal information in the working memory. However, because magnitude information is task-irrelevant in perceptual tasks and not necessarily activated in the working memory, we predict that the SNARC effect here is not elicited by magnitude activated in the working memory. We discussed this hypothesis in General Discussion.

p. 21, l. 526-534

Second, for parity judgment tasks, the SNARC effect is modulated by the relative magnitudes within the tested interval [4]. Moreover, the working memory account posits that the SNARC effect is based on representations created in the serial order working memory [41, 42]. For instance, van Dijck and Fias [42] asked participants to hold a randomly ordered number sequence in the working memory during a parity task, and found a SNARC effect for ordinal positions of the number sequence instead of a SNARC effect for absolute magnitudes. However, here in the perceptual tasks, since the semantic information of magnitude is not necessarily activated in the working memory to solve the task, we hypothesized that the SNARC effect might not be affected by different intervals, but modulated only by the absolute magnitude. Future studies are needed to directly test these predictions. 

A more nuanced discussion of what the authors mean by “automatic” would strengthen the paper considerably.

Response: We agree that a more nuanced discussion of automaticity is important to this paper. According to Tzelgov and Ganor-Stern (2005), automatic processes can be further distinguished as intentional automatic processing, where the process has to be part of the task requirements (e.g., SNARC effect observed in magnitude comparison tasks), and autonomous automatic processing, where the process occurs even when it is not part of the task requirements (e.g., SNARC effect observed in perceptual judgment tasks). Therefore, a stronger examination of the automaticity of numerical-spatial associations would be using perceptual judgment tasks. In the current study, we proposed a series of orientation judgment tasks to specifically focus on how automatic spatial-numerical associations are during autonomous automatic processing.

We have revised our introduction and also throughout the paper to include this distinguish.

p. 3-4, l. 76-84

Using these different kinds of tasks (e.g., parity judgment, magnitude comparison, and perceptual judgment tasks), researchers are able to investigate the extent of automatic processing of numbers in a more nuanced way. Investigating automaticity helps us better understand the internal representations of numbers. According to Tzelgov and Ganor-Stern [14], automatic processes can be further distinguished as intentional automatic processing, where the process has to be part of the task requirements (e.g., SNARC effect observed in magnitude comparison tasks), and autonomous automatic processing, where the process occurs even when it is not part of the task requirements (e.g., SNARC effect observed in perceptual judgment tasks) [14 - 16]. Therefore, a stronger examination of the automaticity of numerical-spatial associations would be using perceptual judgment tasks.

There was another recent paper that examined the automaticity of SNARC effects for perceptual judgment (in this case, color decision), and the authors might find it interesting as the pattern of results they report has some interesting parallels with the current paper (this paper reports that the SNARC effect does not arise for simple color decision, but does arise when participants either perform a go/no-go task, or view the digit for sufficient time before making their decision). The paper may be useful for the authors, particularly for the General Discussion (lines 394-404) when they talk about the need for non-semantic perceptual tasks. The findings are also consistent with one of the authors’ predictions (line 422), where they predict that perceptual judgment tasks might be sensitive to notation (in this paper, non-symbolic numerosities do not show the same pattern of effects as digits):

Cleland & Bull (2019). Automaticity of access to numerical magnitude and its spatial associations: the role of task and number representation, Journal of Experimental Psychology: Learning, Memory and Cognition, 45(2), 333-348.

Response: Thank you so much for letting us know about this paper. We find it very cool and inspiring to investigate the effect of the viewing time of task-irrelevant digit information on the SNARC effect in this paper. We have added this line of research in our introduction and discussion.

p. 4-5, l. 102-113

Activation of magnitude information might also influence the SNARC effect through the response time. Wood and his colleagues [20] did a quantitative meta-analysis of 46 studies with a total of 106 experiments differing in many aspects such as task, population, stimulus modality, stimulus format, and response modality. They found that the longer it took to respond, the larger the SNARC effect was. Similarly, Gevers and colleagues [21] compared the SNARC effect observed from different reaction time bins using a parity judgment task and found that the SNARC effect became stronger with increasing reaction time. More direct manipulation of digit viewing time showed that, contrary to the results that there is a lack of numerical-spatial associations in color decision tasks [11, 12], there appeared to be a SNARC effect in color decision tasks if digits are presented in black shortly (e.g., for 200ms) before color onsets (e.g. blue or green) [9, 10]. In this setting, participants had more time processing digit information and might activate strong enough magnitude information to interfere with reaction time for task-relevant color decisions.

p. 19-20, l. 479-492

However, there are two limitations in the previous studies. First, most studies considered the effect of response time were based on comparisons across different studies or participants, for example, tasks or participants with longer response latencies are associated with larger SNARC effect, which are subjects to sample biases. Second, models explaining the effect of response time on the size of the SNARC effect mainly focus on semantic tasks (e.g, magnitude and parity judgment tasks) that involve working memory. However, a process (e.g., numerical-spatial associations) is more automatic if it can happen when it is not part of the task requirements [14], as in non-semantic perceptual judgment tasks (e.g., orientation judgment tasks). For example, Cleland and Bull [9] found that there is no SNARC effect in a color decision task (whether a digit was blue or green), however, a SNARC effect appeared when the digit was presented in black shortly (e.g., 200 ms) before the color onset. Their findings indicated a stronger SNARC effect as the viewing time of a digit increases. In other words, magnitude information is more likely to interfere with response time when the onset of task-irrelevant magnitude information is earlier than the onset of task-relevant color information, therefore supporting the dual-route account in autonomous automatic processing. 

p. 21, l. 518-525

Because of the differences in perceptual and parity judgment tasks, we made two predictions for perceptual judgment tasks to be further tested. First, for parity judgment tasks, research has shown that the SNARC effect is notation-independent [39]. However, for perceptual judgment tasks, converging evidence suggests the SNARC effect depends on the modalities of stimuli. For example, researchers observed numerical-spatial associations with Arabic digits (i.e., 8), but not with non-symbolic numerosities (e.g., 8 circles), verbal words (e.g., eight), or auditory words (e.g., the sound of eight) [6, 9, 17, 35, 40]. Thus, we predicted that the SNARC effect for perceptual judgment tasks might be more sensitive to the notation of the magnitude than parity judgment tasks. 

I found it difficult to follow the rationale for including Experiment 1. I’m assuming the purpose is to first replicate the finding that Chinese speakers show a left-to-right SNARC effect for parity decision to digits before going onto the more perceptual tasks. However, this rationale needs to be explained in more detail in the Introduction, and the finding should be discussed in the General Discussion.

Response: The main reason for conducting Experiment 1 is to compare the SNARC effect during intentionally automatic processes and autonomous automatic processes. Moreover, we also aimed to replicate previous studies with Chinese speakers. We observed a stronger SNARC effect in the parity judgment task compared to the perceptual task, indicating that the activation strength of magnitude information is stronger in intentionally automatic processes than autonomous automatic processes.

We have added rationales for Experiment 1 in our introduction and further discussed our results in General Discussion.

p. 21, l. 144-151

More specifically, in Experiment 1, we used a parity task to 1) provide a point of comparison for the SNARC effect in intentionally automatic processes. In the parity judgment task, the task-relevant parity judgment might itself activate magnitude information, and 2) replicate the SNARC effect in Chinese-speaking participants [24, 25]. Previous studies have demonstrated the SNARC effect with Chinese-speaking participants. Regardless of whether the participants were readers of predominantly vertical texts (from top to bottom) [24] or readers of predominantly horizontal texts (from left to right) [25], they showed mappings of Arabic numbers onto a horizontal left-to-right number line. 

p. 13, l. 319-321

Furthermore, the SNARC effect elicited in the numeral orientation task was weaker compared to the parity task, indicating that the activation strength of magnitude information is stronger in intentionally automatic processes than autonomous automatic processes.

p. 20, l. 500-503

Moreover, we observed that the size of the SNARC effect is in general larger in the parity judgment task (Experiment 1) than perceptual judgment tasks (Experiment 2), supporting the account that the SNARC effect depends on the activation of magnitude information.

I have outlined further comments by line number below.

Line 66, [9] Keus et al. (2005) is cited as an example of a study that used color naming; however Keus et al. is an ERP study using parity decision so I think there is an error here. I am not aware of any study that uses color *naming* as a task, although there are several that use color decision although not many report a SNARC effect. The authors could cite Fias et al. (2001) or Lammertyn et al. (2002), although neither of these studies found a SNARC effect for color decision. Hoffmann et al. (2013) reported a SNARC effect for children with color decision when the digit was presented in black for 200 ms before changing color. Cleland and Bull (2019) reported a SNARC effect for color decision in adults, but only under certain conditions (see comment above).

Hoffmann, D., Hornung, C., Martin, R., & Schiltz, C. (2013). Developing number–space associations: SNARC effects using a color discrimination task in 5-year-olds. Journal of Experimental Child Psychology, 116(4), 775-791.

Response: We apologized for our mistakes and we have corrected the citations.

p. 3, l. 65-67

Other researchers have investigated automatic numerical-spatial associations using non-semantic tasks, such as phoneme monitoring [6], color judgment [9, 10], orientation judgment [11, 12], and free viewing tasks [13].

9. Cleland AA, Bull R. Automaticity of access to numerical magnitude and its spatial associations: The role of task and number representation. J Exp Psychol Learn Mem Cogn. 2019;45(2):333–48. 

10. Hoffmann D, Hornung C, Martin R, Schiltz C. Developing number-space associations: SNARC effects using a color discrimination task in 5-year-olds. J Exp Child Psychol. 2013;116(4):775–91.

11. Fias W, Lauwereyns J, Lammertyn J. Irrelevant digits affect feature-based attention depending on the overlap of neural circuits. Cognitive Brain Research. 2001;12(3):415-23.

12. Lammertyn J, Fias W, Lauwereyns J. Semantic influences on feature-based attention due to overlap of neural circuits. Cortex: A Journal Devoted to the Study of the Nervous System and Behavior. 2002.

13. Fischer MH, Castel AD, Dodd MD, Pratt J. Perceiving numbers causes spatial shifts of attention. Nature neuroscience. 2003;6(6):555.

Line 110 – the participants are university students so I assume they are used to working with Arabic digits, but it may just be worth clarifying this to the reader (who may be wondering about their proficiency with Arabic digits)

Response: We agree. Our participants are used to working with Arabic digits and have high proficiency with Arabic digits. We clarified this point in our methods.

e.g., p. 7, l. 172

All participants have sufficient experience with Arabic digits.

Line 159 – talk the reader through how you ran the Fias et al. analysis. Was this based on binned responses as well?

Response: We are sorry for not being clear. Regression analysis following Fias et al’s analysis was based on binned responses in our initial submission. We have changed our analysis to regression on individual digits based on your comments below. 

We have clarified our data analysis methods in the manuscript.

p. 8, l. 198-207

Over the past decades, the classical way of analyzing the SNARC effect were regression analysis methods [6, 27]. Individual RT differences (dRT) for each number was computed by subtracting mean RT of left-sided responses from mean RT of right-sided responses. Then the regression analysis of dRT on magnitude of individual numbers would be conducted to measure the size of the SNARC effect. The negative regression slopes indicate the SNARC effect in the expected direction, i.e., faster left-sided (right-sided) responses for small (large) numbers. However, criticism of only using regression analysis to measure the SNARC effect was that although slopes reflect the linear relation between numbers and dRT, the effect size cannot be estimated in terms of proportion of variance explained [28, 29]. Thus, a repeated measures ANOVA of dRT with magnitude as a within-subject factor was suggested by Pinhas and Tzelgov and colleagues [28-30]. 

Line 205 – I can follow the explanation for binning based on the MARC effect in Experiment 1, but do you still need to do this for the orientation tasks? As the participants are not performing a parity task, I don’t think you would expect to see a MARC effect. In particular, I can’t tell from the text whether you have binned the responses for the Fias et al. style analysis, but I’m not sure there’s a reason to do this if you have.

Response: We have changed our ANOVA and regression analysis to individual digits in Experient 2-4.

Experiment 2: p. 11-12, l. 277-298

We conducted a 3 (difficulty: Hard, Medium, Easy) * 10 (magnitude: 0, 1, 2, 3, 4, 5, 6, 7, 8, 9] repeated measures ANOVA of dRT with difficulty and magnitude as within-subject factors. The main effect of magnitude was significant, F(9,279) = 39.71, p < .001, ηp2 =.562, associated with a significant linear trend, F(1,31) = 47.78, p < .001, ηp2 =.606, which indicates an overall SNARC effect. The significant interaction effect between difficulty and magnitude confirmed our hypothesis that the SNARC effect would differ by difficulty level, F(18, 558) = 13.17, p < .001, ηp2=.298. Evaluating different difficulty levels separately, there was a significant SNARC effect in all levels of difficulty (Hard condition: main effect of magnitude F(9,279) = 32.41, p < .001, ηp2 = .511; associated linear trend F(1,31) = 20.75, p < .001, ηp2 = .401; Medium condition: main effect of magnitude F(9,279) = 22.17, p < .001, ηp2 = .417; associated linear trend F(1,31) = 35.82, p < .001, ηp2 = .536; Easy condition: main effect of magnitude F(9,279) = 3.01, p = .002, ηp2 = .088; associated linear trend F(1,31) = 4.92, p = .034, ηp2 = .137).

We additionally analyzed our data following Fias et al.’s [6] method to allow for comparisons with previously published SNARC studies. The regression analysis of individual digits on dRT revealed significant negative slopes (unstandardized) for all levels of difficulty (Fig 2). The Hard condition: B = –6.21, one-tailed comparison of B with zero, t(31) = –4.56, p < .001; the Medium condition: B = –5.31, t(31) = – 5.99, p < .001; the Easy condition: B = –1.58, t(31) = –2.22, p = .017. Furthermore, regression slopes differed across the three levels of difficulty, F(2,62) = 6.18, p = .004, η2 = .166. Pairwise comparisons revealed that regression slopes in the Hard condition were significantly more negative than those in the Easy condition, p = .007; regression slopes in the Medium condition were significantly more negative than those in the Easy condition, p = .009; but the regression slopes were not different between the Hard and Medium conditions.

Experiment 3: p. 15, l. 362-367

Mean dRTs were subjected to repeated measures ANOVA with magnitude (1, 2, 3, 4, 5, 6, 7, 8, 9) as a within-subject factor. The main effect of magnitude was significant, F(8, 280) = 2.72, p = .007, ηp2 =.072, however, the associated linear trend was not significant, F(1, 35) = 1.31, p = .260, indicating the absence of the SNARC effect. The regression analysis of dRT on digits revealed that the slopes (unstandardized) were not significantly different from zero, B = –.94, one-tailed comparison of B with zero, t(35) = –1.15, p = .130 (Fig 4). 

Experiment 4: p. 17-18, l. 414-435

We then computed dRT (RTLeft - RTRight) for each participant and each magnitude. A 3 (difficulty: Hard, Medium, Easy) * 6 (magnitude: 1, 2, 3, 4, 5, 6) repeated measures ANOVA on dRT with difficulty and magnitude as within-subject factors revealed a significant main effect of magnitude: F(5,90) = 7.31, p < .001, ηp2 = .289. Trend analysis revealed a significant overall linear trend, F(1,18) = 28.36, p < .001, ηp2 = .612, indicating an overall SNARC effect. The significant interaction effect between magnitude and difficulty level confirmed our hypothesis that the SNARC effect would differ by task difficulty level, F(10,180) = 2.79, p = .003, ηp2 = .134. Separately analyzing the SNARC effect for each difficulty level, we observed significant SNARC effects in the Hard condition, but not the Medium and Easy conditions (Hard: the main effect of magnitude was significant, F(5,90) = 8.91, p < .001, ηp2 = .331, associated linear trend F(1,18) = 22.35, p < .001, ηp2 = .554; Medium: the main effect of magnitude was not significant, F(5,90) = 1.95, p = .094, but the associated linear trend was significant, F(1,18) = 8.81, p = .008, ηp2 = .329; Easy: the main effect of magnitude was not significant, F(5,90) = .79, p =.559, associated linear trend was not significant, F(1,18) = 2.89, p = .106.

The regression analysis of dRT on digits following Fias et al. [6] revealed significant negative slopes in Hard and Medium conditions (Hard: B = –12.50, one tailed comparison of B with zero, t(18) = –4.73, p < .001; Medium: B = –5.60, t(18) = –2.97, p = .004), but not in the Easy condition, B = –2.02, t(18) = –1.70, p = .053. Moreover, the slopes differed across three difficulty levels, F(2,36) = 7.88, p = .001, ηp2 = .304. Pairwise comparison revealed that the regression slopes in the Hard condition were significantly more negative than those in the Medium (p = .075) and Easy conditions (p = .006), but the regression slopes were not significantly different between the Medium and Easy conditions (p = .403, see Fig 5).

Line 216 – is there a reason not to report exact p-values? Unless it is journal policy, I’d recommend following APA guidelines and reporting exact p-values rather than <.01 or .05.

Response: We agree that reporting exact p-values is better. We have revised our reports to report exact p-values.

Line 277 – I’ve been trying to think through whether it matters that 1 is its own bin whereas all other bins have 2 numbers in them. I am not sure that this is a good idea – would it not make sense to abandon the bins here, given that you cannot have equal numbers of trials in each? Excluding 5 from your stimuli would have been one solution to this.

Response: We changed our analysis to ANOVA with individual digits (1, 2, 3, 4, 5, 6, 7, 8, 9) as a within-subject factor for Experiment 3.

Experiment 3: p. 15, l. 362-367

Mean dRTs were subjected to repeated measures ANOVA with magnitude (1, 2, 3, 4, 5, 6, 7, 8, 9) as a within-subject factor. The main effect of magnitude was significant, F(8, 280) = 2.72, p = .007, ηp2 =.072, however, the associated linear trend was not significant, F(1, 35) = 1.31, p = .260, indicating the absence of the SNARC effect. The regression analysis of dRT on digits revealed that the slopes (unstandardized) were not significantly different from zero, B = –.94, one-tailed comparison of B with zero, t(35) = –1.15, p = .130 (Fig 4). 

Line 289 onwards – I’m uncomfortable with the separate analysis of ranges 1-6 and 4-9. I can see no particular reason why you would predict that there would be a SNARC effect for 1-6 that then reversed for 4-9 (and I note that 4, 5, and 6 are included in both analyses). So why would you run this analysis? I can’t think of a better way of saying it than that this feels like a fishing expedition. There are many ways that you could have sliced up the data, and (unless you have pre-registered this somewhere) I don’t think there’s sufficient justification for this. This is why I've put "no" to the question about whether the analysis is rigorous.

Futhermore, if you are arguing that SNARC effects reverse for the higher number range, then this is a strong claim and needs to be returned to in the General Discussion and (potentially) replicated.

Response: We deleted the separate analysis of ranges 1-6 and 4-9.

Line 304 – “discovered opposing SNARC effects for two number ranges” – I really don’t think you can say this. Firstly, you have sliced up the data without planning to originally. But also, I don’t think you can argue that you have two SNARC effects – the evidence just isn’t strong enough.

Response: We agree that the evidence is not strong enough to reach a conclusion that there is opposing SNARC effects for two number ranges. We have deleted this argument and will further replicate our results to examine this argument in the future.

Line 313 – why are there so many fewer participants in this study than in the previous studies?

Response: To decide the participant number in experiment 4, we did a power analysis based on results of experiment 2 which had a similar design with experiment 4, and found that with power = .8, alpha = .05, we need 5 participants to observe a significant overall SNARC effect. Also, in previous studies with a similar design (e.g., color/orientation decision tasks), the participant number is usually 20. Therefore, we recruited 20 participants in experiment 4. 

Reviewer #2: Overall Evaluation:

The paper is well written and I believe that the experiments operationalize very well the concepts that the authors present in the introduction. The experiments feel in very well a gap of knowledge that the discipline had, moreover confirming Gevers et al.’s (2006) model. I have a couple remarks before I can recommend the manuscript to be accepted. The remarks are listed below. The only main point is that the authors did note completely discuss the fact that they found a SNARC effect only for numbers that go from 1 to 5 in Experiment 3 (then replicated in Experiment 4 with the interval 1-6) and that they found a reverse SNARC effect for numbers that go from 6 to 9 in Experiment 3. My recommendation is to accept the manuscript with minor revision.

Line-by-line comment:

p. 4, l.93

I think “are” is missing in the middle of “which subject”

Response: We revised this sentence.

p. 6, l. 133-135

However, most previous studies that explored the effect of activation of magnitude information through response time were based on either comparison across different studies, different tasks [20] or different participants [21], which are subject to sample biases

p. 5, l.107

The authors write: “The results also provided a point of comparison for the new task of orientation judgment for Experiments 2, 3, and 4.” However, the authors never compare the other experiments to experiment 1, so I am not sure it is really the purpose. I think that a better point of comparison, would have been an experiment with empty squares that are tilted clockwise or counterclockwise (I am not asking for the addition of a supplementary experiment)

Response: We agree and compared the SNARC effect in Experiment 1 and 2 because we used the same number range in these two experiments. The SNARC effect in the parity task (Experiment 1) is in general larger than the SNARC effect in the orientation task (Experiment 2). We added results and discussion about this.

p. 12, l. 302-307

Finally, to compare the size of the SNARC effect in the parity judgment task (Experiment 1) and the numeral orientation judgment task (Experiment 2), we conducted a two-sample t-test between the regression slopes in Experiment 1 and the average regression slopes across three difficulty levels in Experiment 2. Results showed that slopes in Experiment 1 are significantly more negative than the slopes in Experiment 2, t(36.9) = 2.39, p = .01, indicating a stronger SNARC effect in a parity judgment task than a perceptual judgment task.

p. 13, l. 319-321

Furthermore, the SNARC effect elicited in the numeral orientation task was weaker compared to the parity task, indicating that the activation strength of magnitude information is stronger in intentionally automatic processes than autonomous automatic processes.

p. 20, l. 500-503

Moreover, we observed that the size of the SNARC effect is in general larger in the parity judgment task (Experiment 1) than perceptual judgment tasks (Experiment 2), supporting the account that the SNARC effect depends on the activation of magnitude information.

p. 6, l.148

I would like to know on what ground the authors determined a cut-off at 1500ms?

Response: We used 1000ms as reaction time criterion for our Experiment 2 – 4 (i.e., orientation tasks) literature to exclude slow outliers because it is a commonly used criterion in orientation judgment tasks in the SNARC effect (e.g., Fias, Lauwereyns, & Lammertyn, 2001; Mitchell, Bull, & Cleland, 2012). We increased this criterion to 1500ms for our Experiment 1(i.e., a parity task) because we thought it might be more difficult and take longer to make a parity judgement than orientation judgment (as indicated in our results, Experiment 1: mean RT = 510 ms; Experiment 2: mean RT = 474 ms; Experiment 3: mean RT = 483 ms; Experiment 4: mean RT = 448 ms).

p. 6, l.178

Why did the authors use 37 participants, what was the rational in terms of power of the analysis? I am asking because in the first experiment only 32 participants were used whereas in experiment 4, 20 were used.

Response: In previous studies with a similar design (e.g., parity/color/orientation decision tasks), the participant number is usually 20 (Fias, 2001; Fias et al., 2001; Lammertyn et al., 2002). For Experiment 1-3, we recruited more than 20 participants to reach enough power after potentially deleting some participants that might randomly press buttons to get paid.

To decide the participant number in experiment 4, we did a power analysis based on results of experiment 2 which had a similar design with experiment 4, and found that with power = .8, alpha = .05, we need 5 participants to observe a significant overall SNARC effect. Thus, we recruited the same number of participants as previous studies ((Fias, 2001; Fias et al., 2001; Lammertyn et al., 2002).

p. 10, l.241

The authors write: “However, there was a potential confound in this task design. The perceptual characters of each Arabic digit might have led to different levels of difficulty, as indicated by the significant main effect of number on RT, F(9, 279) = 7.10, p < .001, ηp2 = .186. Experiment 3 overcame this problem with a modified perceptual judgment task.” Could they be more explicit, I am asking this because the digits (and therefore their perceptual characters) are manipulated orthogonally to the task difficulty, so I don’t see how there could be a confound?

Response: We apologize for not being clear. We meant sub-levels of difficulty for different digits. For instance, the rotated digit “1” could be easier to define its orientation than “3” in the same rotational degree because the rotation status of straight lines might be easier to clarify than that of curved lines, thus helping the overall performance of “1” over “3”. Because the SNARC effect is based on an association between small (large) numbers and left-side (right-side) response, different sub-levels of difficulty for different digit might confound the SNARC effect.

p. 13, l. 322-328

However, there was a potential confound in this task design. The perceptual characters of each Arabic digit might have led to different sub-levels of difficulty for different digits, as indicated by the significant main effect of number on RT, F(9, 279) = 7.10, p < .001, ηp2 = .186. For instance, the rotated digit 1 could be easier to define its orientation than 3 in the same rotational degree because the rotation status of straight lines might be easier to clarify than that of curved lines, thus helping the overall performance of 1 over 3. Experiment 3 overcame this problem with a modified perceptual judgment task.

p. 11, l.273

Why did the authors cut reaction times over 1000ms here (same in experiment 2) while cutting reaction times over 1500ms in experiment 1?

Response:

We used 1000ms as reaction time criterion for our experiment 2 – 4 (i.e., orientation tasks) literature to exclude slow outliers because it is a commonly used criterion in orientation judgment tasks in the SNARC effect (e.g., Fias, Lauwereyns, & Lammertyn, 2001; Mitchell, Bull, & Cleland, 2012). We increased this criterion to 1500ms for our experiment 1(i.e., a parity task) because we thought it might be more difficult and take longer to make a parity judgement than orientation judgment (as indicated in our results, Experiment 1: mean RT = 510 ms; Experiment 2: mean RT = 474 ms; Experiment 3: mean RT = 483 ms; Experiment 4: mean RT = 448 ms).

General Discussion

The general discussion is good but it does not seem (or maybe I missed it) to address the elephant in the room. Why is there a SNARC effect only for numbers that go from 1 to 5 in Experiment 3 and then replicated in Experiment 4? And why there seem to be a reverse SNARC effect for numbers that go from 6 to 9 in Experiment 3. The authors would need to address that.

Response: We think that the stronger SNARC effect in a small number range (1-6) compared to the whole number range (1-9) might indicate noise in automatic representations of numbers. More specifically, the noise in representation of numbers increases as number increases, causing an uneven distribution of numerical-spatial associations. Previous studies using non-symbolic numerosities (1-9 circles) in an orientation decision task also revealed a stronger SNARC in 1-4 compared to 6-9, indicated a more precision spatial association in a smaller number range (Mitchell, Bull, & Cleland, 2012). Therefore, a potential explanation of our results is that there might be more noise in representations and a less associated spatial precision in the larger range when magnitude information is weakly activated.

We deleted the separate analysis of 1-6 and 4-9 number range in Experiment 3 as a reversed SNARC effect might be a too strong argument based on current results suggested by our reviewer. 

We have revised our discussion as follows.

p. 22, l. 535-551

Furthermore, the automatic numerical-spatial associations might provide an insight into the representation of magnitude. It is generally believed that the representation of nonsymbolic numerosities (e.g., 20 apples) becomes noisier as the number increases in an Approximate Number System (ANS) [43]. This is suggested by two main accounts: the linear model (linear representations of numbers with linearly increasing variability as magnitude increases) [44] and the log model (logarithmic representations of numbers with fixed variability around numbers) [45]. Consistent with these accounts, previous studies using non-symbolic numerosities (1-9 circles) in an orientation decision task also revealed a stronger SNARC in 1-4 compared to 6-9, indicated a more precision spatial association in a smaller number range [35].

As for Arabic digits, educated adults are able to represent numbers linearly [46]. A common task to measure the representation of numbers is the number line task [46] (e.g., where is 345 on a 0-1000 number line?), where intentional processing of magnitude information is required. However, in the current study, we observed potentially stronger spatial-numerical associations in a smaller number range (1-6) than a larger number range (1-9) using a task where magnitude information is task-irrelevant (Experiment 3 and Experiment 4). A potential explanation is that there might be more noise in representations and a less associated spatial precision in the larger range when magnitude information is weakly activated.

---

## [Decision Letter · Decision Letter 1]

23 Jan 2020

PONE-D-19-16766R1

Automaticity in processing spatial-numerical associations: Evidence from a perceptual orientation judgment task of Arabic digits in frames

PLOS ONE

Dear Dr. Zhang,

Thank you for submitting your manuscript to PLOS ONE. After careful consideration, we feel that it has merit but does not fully meet PLOS ONE’s publication criteria as it currently stands. Therefore, we invite you to submit a revised version of the manuscript that addresses the points raised during the review process.

As you will see, both reviewers suggest publication of this revised manuscript. I agree with the reviewers: you did a very good job in responding to the reviewers and that this manuscript will make a nice contribution to the field. Reviewer 1 suggests a minor revision that I invite you to consider. So, I am sending the manuscript back to you with ‘minor revision’. Once you have double checked the issue raised by the reviewer, please re-submit the manuscript.

We would appreciate receiving your revised manuscript by Mar 08 2020 11:59PM. To enhance the reproducibility of your results, we recommend that if applicable you deposit your laboratory protocols in protocols.io, where a protocol can be assigned its own identifier (DOI) such that it can be cited independently in the future. For instructions see: http://journals.plos.org/plosone/s/submission-guidelines#loc-laboratory-protocols

We look forward to receiving your revised manuscript.

Kind regards,

Claudio Mulatti, Ph.D.

Academic Editor

PLOS ONE

Reviewers' comments:

Reviewer's Responses to Questions

**Comments to the Author**

1. If the authors have adequately addressed your comments raised in a previous round of review and you feel that this manuscript is now acceptable for publication, you may indicate that here to bypass the “Comments to the Author” section, enter your conflict of interest statement in the “Confidential to Editor” section, and submit your "Accept" recommendation.

Reviewer #1: (No Response)

Reviewer #2: All comments have been addressed

2. Is the manuscript technically sound, and do the data support the conclusions?

Reviewer #1: Yes

Reviewer #2: Yes

3. Has the statistical analysis been performed appropriately and rigorously? 

Reviewer #1: Yes

Reviewer #2: Yes

4. Have the authors made all data underlying the findings in their manuscript fully available?

Reviewer #1: Yes

Reviewer #2: Yes

5. Is the manuscript presented in an intelligible fashion and written in standard English?

Reviewer #1: Yes

Reviewer #2: Yes

6. Review Comments to the Author

Reviewer #1: I'd like to thank the authors for their thorough and well-written cover letter. They have done a good job of responding to the reviewers' comments and I wish them luck with this line of research.

I am happy to see the work published as it is now - I just have one very minor point of clarification. I think line 541 (the version without track changes) has a minor error. I think that "non-symbolic numerosities (1-9 circles) in an orientation decision task" should read "non-symbolic numerosities (1-9 triangles) in an orientation decision task". The paper cited does use circles for color decision, but I believe the orientation task used triangles - perhaps the authors could double-check this.

Reviewer #2: I commend the authors as they have answered all my concerns, I think that the manuscript makes a nice contribution.

7. PLOS authors have the option to publish the peer review history of their article (what does this mean?). If published, this will include your full peer review and any attached files.

Reviewer #1: No

Reviewer #2: No

---

## [Author Response · Author response to Decision Letter 1]

30 Jan 2020

Review Comments to the Author

Reviewer #1: I'd like to thank the authors for their thorough and well-written cover letter. They have done a good job of responding to the reviewers' comments and I wish them luck with this line of research.

I am happy to see the work published as it is now - I just have one very minor point of clarification. I think line 541 (the version without track changes) has a minor error. I think that "non-symbolic numerosities (1-9 circles) in an orientation decision task" should read "non-symbolic numerosities (1-9 triangles) in an orientation decision task". The paper cited does use circles for color decision, but I believe the orientation task used triangles - perhaps the authors could double-check this.

Response: Thank you so much for your comments and correction. We corrected this mistake.

p. 22, l. 540 - 543:

Consistent with these accounts, previous studies using non-symbolic numerosities (1-9 triangles) in an orientation decision task also revealed a stronger SNARC in 1-4 compared to 6-9, indicated a more precise spatial association in a smaller number range [35].

Reviewer #2: I commend the authors as they have answered all my concerns, I think that the manuscript makes a nice contribution.

Response: Thank you so much for your comments.

---

## [Editor Report · Decision Letter 2]

31 Jan 2020

Automaticity in processing spatial-numerical associations: Evidence from a perceptual orientation judgment task of Arabic digits in frames

PONE-D-19-16766R2

Dear Dr. Zhang,

We are pleased to inform you that your manuscript has been judged scientifically suitable for publication and will be formally accepted for publication once it complies with all outstanding technical requirements.

With kind regards,

Claudio Mulatti, Ph.D.

Academic Editor

PLOS ONE
---

## [Editor Report · Acceptance letter]

5 Feb 2020

PONE-D-19-16766R2 

Automaticity in processing spatial-numerical associations: Evidence from a perceptual orientation judgment task of Arabic digits in frames 

Dear Dr. Zhang:

I am pleased to inform you that your manuscript has been deemed suitable for publication in PLOS ONE. Congratulations! Your manuscript is now with our production department. 

With kind regards,

on behalf of

Dr. Claudio Mulatti 

Academic Editor

PLOS ONE